# Recommendations of older adults on how to use the PROM 'TOPICS-MDS' in healthcare conversations: A Delphi study

Ruth E. Pel-Littel [1,2]*, Cynthia S. Hofman[1]*, Liesje Yu[3], Silke F. Metzelthin[4], Franca H. Leeuwis[5], Jeanet W. Blom[6], B. M. Buurman[2], Mirella M. Minkman[1,7]

**1** Department of Innovation and Research, Vilans, Centre of expertise for long-term care, Utrecht, the Netherlands, **2** Department of Internal Medicine, Section of Geriatric Medicine, Amsterdam UMC, University of Amsterdam, Amsterdam, the Netherlands, **3** Faculty of Earth and Life Sciences, VU University, Amsterdam, the Netherlands, **4** Department of Health Services Research, Care and Public Health Research Institute, Maastricht University, Maastricht, the Netherlands, **5** Department of Geriatric Medicine, Radboud University Medical Center, Nijmegen, the Netherlands, **6** Department of Public Health and Primary Care, Leiden University Medical Center, Leiden, the Netherlands, **7** University of Tilburg, TIAS school for Business and Society, Tilburg, the Netherlands

* r.pel@vilans.nl (RPL); C.Hofman@vilans.nl (CH)

**Data Availability Statement:** The minimal anonymized data set of the Delphi study has been

## Abstract

In shared decision making, the exploration of preferred personal health outcomes is important. Patient-reported outcome measures (PROMs) provide input for discussions between patients and healthcare professionals. The Older Persons and Informal Caregivers Survey Minimum DataSet (TOPICS-MDS) PROM is a multidimensional questionnaire on the physical and mental health and wellbeing of older adults. This study investigates how the TOPICS-MDS could be used in individual healthcare conversations. We explored views of older adults regarding 1) whether the health domains they want to discuss are included in the TOPICS-MDS and 2) the comprehensibility of the TOPICS-MDS for healthcare conversations with older adults. A three-round Delphi study was conducted. A total of 57 older adults participated in the study, the mean (SD) age was 71.5 (8.5) years, and 78.9% of the participants were female. The participants were divided into four panels based on educational level and cultural background. We used online questionnaires and focus groups. Consensus was pre-defined to be the point when ≥75% of the participants agreed that a domain was important or very important (scored on a 5-point Likert scale). The inter-expert agreement was computed for Round 1 and 3 with Kendall's W. Round 2 was a focus-group. Qualitative data were analyzed by content analysis. Older adults considered 'functional limitations', 'emotional wellbeing', 'social functioning' and 'quality of life' to be important domains of the TOPICS-MDS to discuss in healthcare conversations. The participants added 'coping with stress', 'dealing with health conditions and the effects on life' as extra domains for healthcare conversations. Challenges regarding the comprehensibility of the TOPICS-MDS included difficult words and lengthy or sensitive questions. Questions that included multiple topics were difficult to understand. The TOPICS-MDS covers the domains of life that older adults value as important to discuss with healthcare professionals, and two additional

uploaded to fig share; https://doi.org/10.6084/m9.figshare.10001906.v1.

**Funding:** This work is supported by ZonMw [grant number 633400019]. The funding agency had no role in the study design, methods, and subject requirement; in the collection, analysis and interpretation of data; in the writing of the report; or in the decision to submit the article for publication. www.zonmw.nl.

**Competing interests:** The authors have declared that no competing interests exist.

domains were identified. For older adults with a low level of education or a culturally diverse background, the TOPICS-MDS needs to be adjusted for comprehensibility.

## Introduction

Shared decision making (SDM) facilitates the communication between healthcare professionals and patients when decisions have to be made about diagnostic procedures, treatment options or care options. For older adults, SDM becomes much more complex when there are multiple chronic conditions (MCC) to take into account. MCC are highly prevalent in older adults [1,2] and are defined as two or more concurrent chronic conditions that collectively might have an adverse effect on one's health status, daily functioning or quality of life [3–7]. The presence of MCC in older adults influences the goals and expectations of care and treatment. For instance, the ability to maintain independence and a good quality of life often becomes more important than life expectancy [8,9]. Therefore, it is important to explore preferred health outcomes first, before focusing on the treatment options for one or more conditions. In other words, it is recommended to treat the patient rather than the disease.

Accordingly, the 'dynamic model of SDM in frail older patients' was developed for older adults with MCC. The model emphasizes the exploration of personal goals and preferred health outcomes as an initial step preceding the deliberation about concrete treatment options [10]. However, reflecting on one's own health and formulating preferences for health outcomes in a discussion with a healthcare professional is not common practice for older adults [11–13]. Older adults often lack the confidence to express their personal views, and healthcare professionals often fail to elicit older adults' preferences and goals [11,13–15].

Patient-reported outcomes measures (PROMs) are used for quality improvement, research and benchmark purposes, but they can also be used as input for individual healthcare conversations between patients and healthcare professionals. Therefore, the use of a PROM could support SDM by providing a tool for older adults to reflect on their health and empower them to discuss topics and health outcomes that are important to them [16]. For this purpose, the Dutch Geriatric Society selected the 'Older Persons and Informal Caregivers Survey Minimum DataSet' (TOPICS-MDS) PROM [17]. The TOPICS-MDS comprises two multidimensional questionnaires: (1) one on the physical and mental health and wellbeing of older adults and (2) one on the caregiver burden and quality of life of informal caregivers. The creation of the TOPICS-MDS was a joint effort by eight Dutch universities participating in the National Care for the Elderly Program. The TOPICS-MDS has been compiled by a working group of experts and is based on validated and reliable questionnaires [18]. However, since the TOPICS-MDS was originally developed for research purposes, this PROM is not often used in individual healthcare conversations. The aim of this study was to explore whether the TOPICS-MDS, the questionnaire for older adults, could facilitate SDM conversations between professionals and older adults with MCC. Therefore, we (1) explored which health domains are important to discuss with healthcare professionals according to older adults in the Netherlands and (2) examined the comprehensibility of the current TOPICS-MDS items. We conducted a modified Delphi study with older adults.

## Methods

### Participants

The Delphi panel in this study comprised older adults living in the Netherlands. This study aimed explicitly to include a diversity of backgrounds (e.g., age, sex, education and cultural

diversity) to reflect the mixed population of older adults in the Netherlands and support the external validity of this study [19]. Participants were recruited through purposive sampling, which means they were recruited based on characteristics of the population and the purpose of the study. The inclusion criteria were persons 1) older than 50 years, 2) with sufficient Dutch language skills to understand the questionnaires in written or spoken form and 3) living in the Netherlands. We deliberately selected the age threshold of ' ≥ 50 years' since research shows that persons with a low level of education or a culturally diverse background suffer from MCC at a relatively young age [20]. Participants with a culturally diverse background or a low level of education were approached with the assistance of Pharos (Dutch Centre of Expertise on Health Disparities). In the Netherlands, there are over 200 different nationalities; therefore, it was impossible to involve all nationalities. Furthermore, it is difficult to recruit people with a low level of education to participate in research. Pharos has a broad network of 'key figures' in the populations that they approached to find eligible participants for the focus groups. This approach guided the final composition of the groups. The participants with a Hindu background originated mostly from the former Dutch colony of Suriname, and the participants with a Moroccan background were immigrant workers; thus, two different types of migrants in the Netherlands were represented. Highly educated (secondary education, college or university) participants were recruited by a website for older adults (Beteroud.nl) and social media (Facebook and Twitter).

## Design of the Delphi study

The Delphi method is defined as 'a group process involving an interaction between the researcher and a group of identified experts on a specific topic' [21]. The Delphi method is widely used and accepted to reach a consensus [21–23]. Delphi methodology combines quantitative and qualitative exploration of subjective assumptions surrounding a given topic and elicits opinions from experts to obtain a group response and consensus [24]. A Delphi study is usually conducted in sequence rounds, which allows the experts to adapt their opinions and facilitates the yielding of consensus [25].

The Delphi procedure used in this study comprised three rounds that were completed over a three-month period in 2017. The first round of the Delphi study aimed to identify the important health domains to discuss in healthcare conversations according to older adults. The second round aimed to gain insights into the comprehensibility of the questions and response options of the TOPICS-MDS. The third round aimed to re-evaluate undecided health domains and reformulate TOPICS-MDS questions and response options.

Within the Delphi procedure, two methods were applied to obtain data within the rounds, namely, an online questionnaire and focus group sessions. These methods were chosen to allow a mixed composition of the panel and the ability to support respondents in their understanding of the questions. Participants with a culturally diverse background or a low level of education were questioned in focus groups during Delphi Rounds 1 and 2. Focus groups allow more interaction between the panel experts and with support from the facilitators that guarantees attention for each participant, thus leading to a better understanding of the TOPICS-MDS when discussed with these specific groups [26]. In Fig 1, an overview is presented of the study aims, methods and characteristics of participants.

## Ethical considerations

The requirement to obtain approval for this study was waived by the Institutional Review Board of the UMCU (19/355). For the online section of this study, participants provided their

| | Aim | Method | Participants |
|---|---|---|---|
| Round 1 | Assess health domains | Online questionnaire<br><br>Focusgroups part 1<br> F1: low education<br> F2: cultural diverse background<br> F3: cultural diverse background | N = 44<br><br>N = 6 Low educ.<br>N = 4 Moroccan<br>N = 3 Hindu |
| Round 2 | Discuss formulation of unclear questions and respons options | Focusgroup<br> F4: high education<br><br>Focusgroups part 2<br> F1: low education<br> F2: cultural diverse background<br> F3: cultural diverse background | N = 5<br><br>N = 6 Low educ.<br>N = 4 Moroccan<br>N = 3 Hindu |
| Round 3 | Re-assess health domains &<br>Assess reformulated questions and response options | Online questionnaire | N = 24 (remaining participants round 1) |

**Fig 1. Design of the Delphi study.**

written informed consent, and the focus group discussion participants provided their verbal consent, which was audio recorded and transcribed verbatim.

### Design of the Delphi questionnaire

The Delphi questionnaire was based on the original TOPICS-MDS (version before 2017) [18] and presented to the participants in both the online questionnaire and the focus group. The TOPICS-MDS comprises 51 questions and collects data in seven domains: demographics, morbidity, quality of life, functional limitations, mental health, social functioning and health service use. From these domains, four health domains from the TOPICS-MDS were evaluated in the Delphi: functional limitations, mental health, social functioning and quality of life. The other three domains, namely, demographics, morbidity and health service utilization, were not included in the Delphi study since those domains are core topics (case mix variables) in relation to health outcomes and thus do not need be evaluated in the Delphi. Since recent literature describes a more holistic scope on health domains, as expressed in the new definition of health from the World Health Organization and the literature about positive health, we added

two more domains to the questionnaire: meaning in life and social and living environments [27–29].

The questionnaire began with questions regarding the older adult's demographic characteristics, followed by an open question about the current wellbeing of the older adult: 'How would you rate your general wellbeing (range 0–10)?' This question was followed by questions about the importance of discussing each health domain with a healthcare professional in order to gain insight into the participant's personal health outcomes. For each domain, the importance was assessed on a 5-point Likert scale (range: 'not important' to 'very important'). The content of the Delphi questionnaire was similar in both the online questionnaire and the focus groups, although in the focus groups, the 5-point Likert scale was expressed in yellow paper stars that the participants could distribute to assess the importance.

**First round.** In Round 1, we aimed to determine the health domains that older adults consider important to discuss in healthcare conversations. Participants with a high level of education received the questionnaire (S1 File Round 1: online questionnaire) through a hyperlink, and participants with a low level of education or a culturally diverse background discussed the questions from the questionnaire during focus groups. For each domain, the importance was assessed on a 5-point Likert scale (range: 'not important' to 'very important'). After assessing each health domain, participants were asked to provide a rationale regarding their answer. At the end, participants could add other health domains that they considered important to discuss with their healthcare professional and indicate the reason(s) for adding the domain(s). In the focus groups, all health domains were discussed plenarily, and participants were asked to assess the importance of the different health domains by means of a visual 5-point scale.

**Second round.** Round 2 was conducted exclusively in focus groups. There were four focus groups: one with participants with a high level of education (recruited from the participants that completed the online survey), one with participants with a low level of education and two with participants with culturally diverse participants. The aim of Round 2 was to gain insights into the comprehensibility of the TOPICS-MDS questions and response options when used in healthcare conversations. From the public TOPICS-MDS data repository, we derived the items from the questionnaire that had > 10% missing entries (n = 12) [18]. We presumed that these twelve questions were difficult for people to answer, but the reason for the missing entries was unknown. These questions were discussed plenarily. Participants were asked whether the formulations were understandable and clear and how the ambiguous questions needed to be revised.

**Third round.** Round 3 was conducted solely through an online questionnaire. This round aimed to assess the (modified) set of health domains and evaluate the reformulated questions and response options. The revised questionnaire was e-mailed to participants from the first round who had agreed to participate in the third round (S2 File. Round 3: online questionnaire). A summary of the comments on the health domains of the first round was shared with the participants. Using a Likert scale ranging from 1 (lowest) to 5 (highest) in levels of importance, the participants were required to assess each remaining health domain on the basis of the question, 'How important is it to you that your healthcare professionals asks you about . . .(health domain)?' In addition, the revised questions and response options from the focus group sessions were evaluated for comprehensibility.

## Data analysis

Descriptive statistics were used to summarize the sociodemographic data of the participants and the quantitative data. The quantitative data were analyzed with 'Statistical Package for the Social Sciences' (SPSS, version 23) software. A consensus was reached when $\geq$ 75% of the participants

assessed a health domain as important (4) or very important (5). When ≤ 50% of the participants assessed a domain as important (4) or very important (5), the domain was excluded. Items with an acceptance rate between 50% and 75% were re-evaluated in the next round. These consensus agreement scores were based on the methods of relevant, recently published Delphi studies [30,31]. The inter-expert agreement was computed for Round 1 and 3 with Kendall's W.

The qualitative data comprised the in-depth explanations of the motivations why participants ranked a domain as important to discuss in healthcare conversations. These data were collected with the purpose of enabling the participants of the following Delphi round to re-rank the health domains of the TOPICS-MDS with the group's viewpoint in mind. The data included open text fragments (online questionnaire) and audio recordings (focus groups). The recordings of the focus groups were transcribed verbatim. First, the transcripts were read multiple times, so the reader was familiarized with the data. Subsequently, the transcripts were analyzed thematically by 'open coding' via the program 'MAXQDA'. Following axial coding, the main themes were clustered and combined. Two researchers (RPL and LY) coded the first two transcripts independently and discussed them to reach a consensus. The remaining transcripts were coded by LY and subsequently discussed with RPL.

The remaining health domains and the restated questions and answer options, including the remarks made by participants in the previous rounds, were presented in Round 3 (S2 File. Round 3: online questionnaire). A consensus on the definitive list of important health domains was reached when ≥ 75% of the participants assessed a health domain as important (4) or very important (5). In addition, the restated questions and response options were presented. When ≥ 75% of the participants evaluated the restated question as better than the original question, it was recommended to carry out the adjustments.

## Results

### Demographics of the participants

The demographic characteristics of the 57 participants are described in Table 1. The mean (SD) age was 71.5 (8.5) years, and 78.9% of the participants were female. Of the participants, 17 (29.8%) had a low level of education (primary school or less), and 13 (22.8%) had a culturally diverse background. These participants were Algerian (N = 1), Belgian (N = 1), Curaçaoan (N = 1), German (N = 1), Scottish (N = 1), Surinamese (N = 4) and Moroccan (N = 4). All participants (N = 57, 100%) lived independently or with relatives.

### Health domains to discuss in healthcare conversations: Quantitative results

In Round 1 (n = 57), we explored the importance of the six health domains to discuss in healthcare conversations according to older adults. As shown in Fig 2, none of the health domains were assessed as important (4) or very important (5) by more than 75% of the participants. Between 50 and 75% of the respondents assessed the original health domains, functional limitations, emotional wellbeing, social functioning and quality of life, as important (4) or very important (5) to discuss in healthcare conversations. Therefore, these domains were re-evaluated in the third round. Less than 50% of the participants assessed the added domains, meaning in life and living and social environments, as important (4) or very important (5) to discuss in healthcare conversations. Consequently, these domains were excluded.

Participants of Round 1 introduced three other domains that they felt were important to discuss with their healthcare professionals and are not domains in the current TOPICS-MDS. These domains were physical exercise (e.g., walking, cycling), coping with stress, and how to deal with the disease and its effects on daily life. These domains were included in Round 3, in

**Table 1. Characteristics of the participants.**

| Demographic characteristics | Participants in the online questionnaire with a high level of education n = 44 | Participants in the focus group with a low level of education n = 6 | Participants in the focus groups with a culturally diverse background n = 7 | Total N = 57 |
|---|---|---|---|---|
| **Sex** (n,%) | | | | |
| Female | 33 (75) | 5(83) | 7 (100) | 45 (78.9) |
| **Age** (mean, SD) | 73.0 (7.5) | 66.7 (11.3) | 65.8 (9.5) | 71.5 (8.5) |
| **Level of education** (n, %) | | | | |
| High | 40 (90.9) | 0 (0) | 0 (0) | 40 (70.2) |
| Low | 4 (9.1) | 6 (100) | 7 (100) | 17 (29.8) |
| **Ethnicity** (n,%) | | | | |
| Dutch | 41 (93.2) | 3 (50) | 0 (0) | 44 (77.2) |
| Other | 3 (6.8) | 3 (50) | 7 (100) | 13 (22.8) |
| **Marital status** (n,%) | | | | |
| Married/living together | 21 (47.7) | 0(0) | 4 (57.1) | 25 (43.9) |
| Single | 23 (52.3) | 6 (100) | 3 (42.9) | 32 (56.1) |
| **Living situation** (n,%) | | | | |
| Independent | 44 (100) | 6 (100) | 7 (100) | 57 (100) |

which the health domains from Round 1 with an acceptance rate between 50% and 75% were re-evaluated.

As depicted in Fig 3, after Round 3 (n = 24), the remaining health domains to discuss in health-care conversations were functional limitations, emotional wellbeing, social functioning, quality of life, coping with stress and dealing with health conditions and the effects on daily life. The majority of the participants did not assess physical exercise as (very) important to discuss in healthcare conversations. The inter-expert agreement (Kendall's W) was for Round 1 0.083 and for Round 3 0.058. Based on these low inter agreement scores we further used descriptive statistics.

## Differences between the subgroups

There are differences between participants with a high level of education (n = 44), participants with a low level of education (n = 6) and participants with a culturally diverse background (n = 7). An overview of how the health domains were rated in Round 1 by the different groups of participants can be found in Fig 4. More than 75% of the participants with a low level of education or a culturally diverse background assessed functional limitations, emotional wellbeing, social functioning and quality of life as important (4) or very important (5), while less than 75% of the participants with a high level of education assessed these domains as important (4) or very important (5).

## Health domains to discuss in healthcare conversations: Qualitative results

Participants in Round 1 were invited to substantiate their assessment of health domains, both in the online questionnaire and in the focus groups. The qualitative data are structured

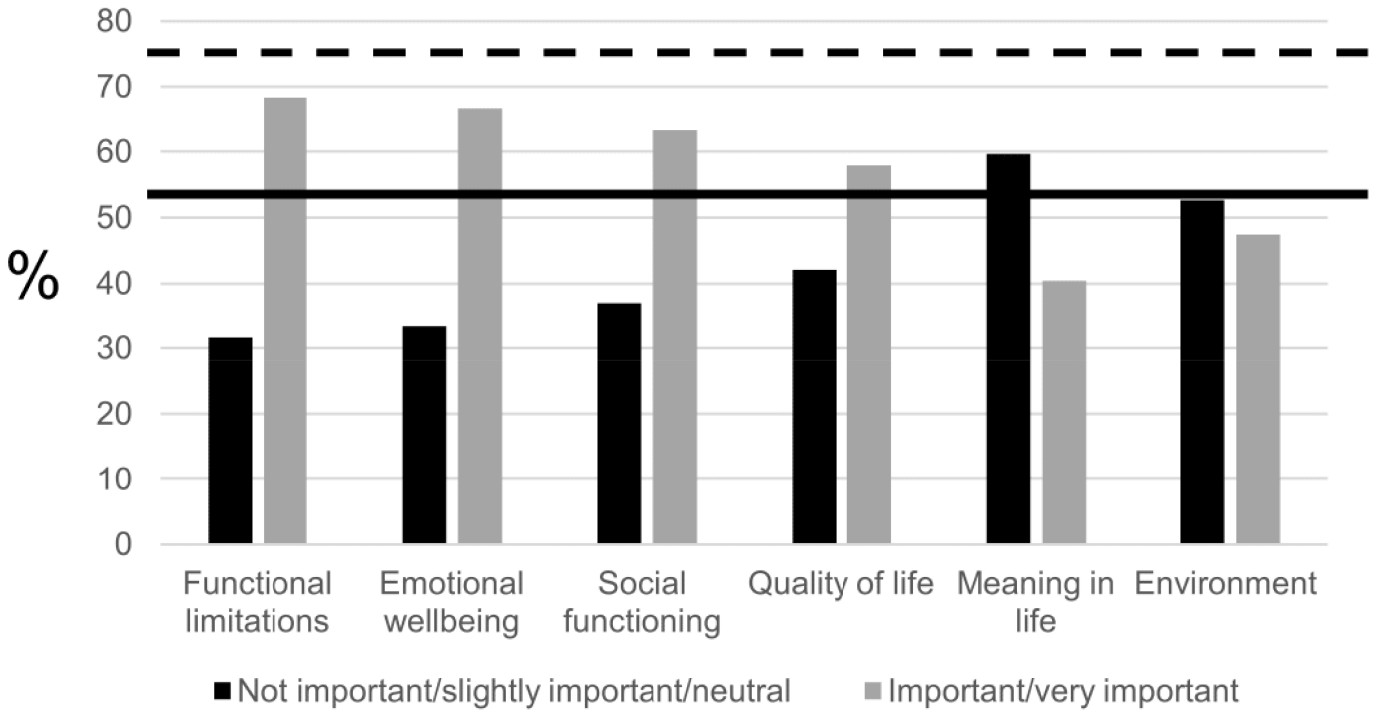

**Fig 2. Round 1: Overview of the assessments of the health domains.** Frequency distribution (%) of the importance to discuss each domain with healthcare professionals. The bold black line represents the 50% threshold. The bold black dotted line represents the 75% threshold.

according to each health domain as described in the TOPICS-MDS: functional limitations, emotional wellbeing, social functioning and quality of life.

**Functional limitations.**   The domain of functional limitations includes basic and instrumental activities of daily life and mobility, such as walking[18]. Some participants wanted to discuss their limitations in daily life with healthcare professionals since they experience physical problems. The participants mentioned that it is important to discuss this topic with healthcare professionals since healthcare professionals can arrange adequate support:

*'I find it important to discuss functional limitations with my healthcare professional since he can arrange extra domestic help.' (female participant, 81 years with a high level of education, online questionnaire).*

**Emotional wellbeing.**   The domain of emotional wellbeing concerns the mental state of a person, e.g. feeling nervous, calm, happy, depressed etc.[18] Some participants found it important to discuss their emotional wellbeing with healthcare professionals because they tend to hide depressing thoughts from others:

*'It is important that healthcare professionals ask about my emotional wellbeing because sometimes I feel depressed and if they ask me how I am doing emotionally, I can talk about it.' (female participant, 87 years, with a high level of education, online questionnaire).*

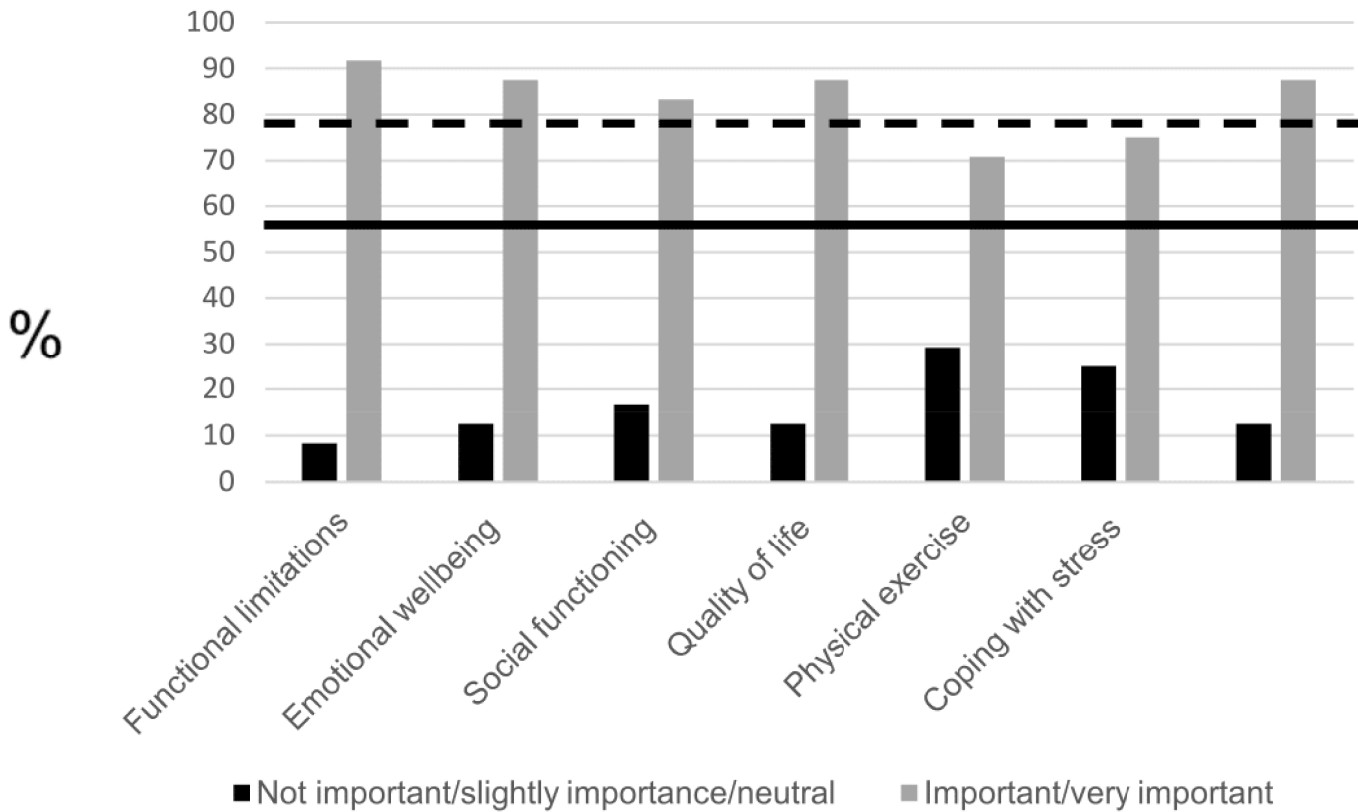

**Fig 3. Round 3: Overview of the remaining health domains.** Frequency distribution (%) of the importance to discuss each domain with healthcare professionals. The bold black line represents the 50% threshold. The bold black dotted line represents the 75% threshold.

Furthermore, some participants indicated that the relationship between the healthcare professional and the client is of importance:

*'Yes, I would like to discuss this topic if I know the healthcare professional. If I have to stay one or two days in the hospital, I don't want to tell them everything. The relationship with my GP (General Practitioner) is very important to me.' (female participant, 71 years in a focus group with a culturally diverse background).*

**Social functioning.** The domain of social functioning within the TOPICS-MDS reflects if a person feels impeded in entering into social contacts[18]. Participants found social functioning an important topic to discuss because many older adults live alone:

*'When you become older and when you live alone, loneliness plays a role.' (female participant, 67 years with a high level of education, online questionnaire).*

However, the participants thought that loneliness is a sensitive topic to discuss. Therefore, if healthcare professionals ask the correct questions, older adults may feel at ease when discussing these topics. Some participants found it important that this topic is discussed with healthcare professionals because the healthcare professionals can provide suggestions or solutions for this problem.

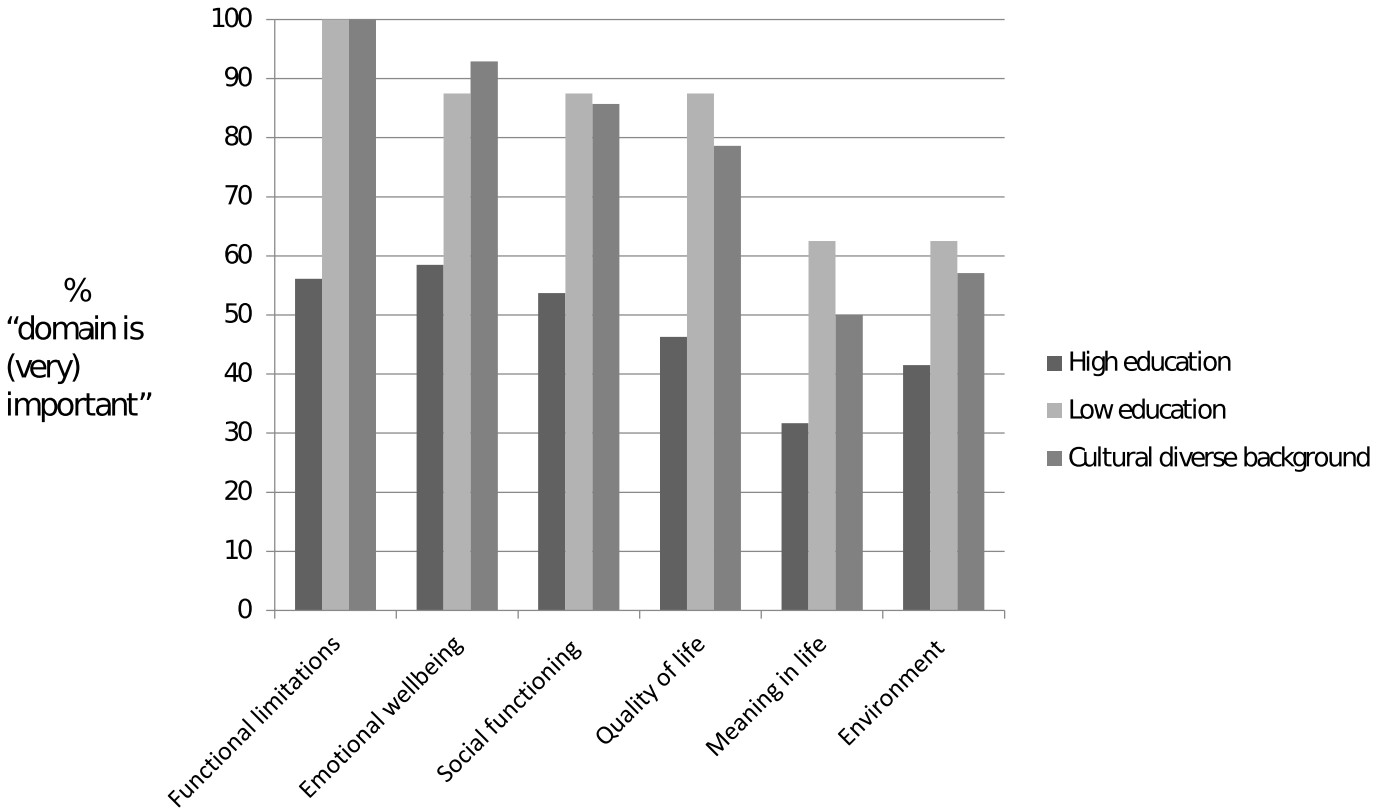

**Fig 4. Overview of the assessments of the health domains by subgroups of participants.** Frequency distribution (%) of the importance of discussing each domain with healthcare professionals. The light gray row represents the highly educated participants, the middle gray row represents the participants with a low level of education and the dark gray row represents participants from culturally diverse backgrounds.

**Quality of life.**   The domain of quality of life is composed of several attributes, such as mobility, self-care, usual activities, pain/discomfort, anxiety/depression and cognitive function [18]. Some participants indicated that the focus should not be on the different chronic conditions but on how these conditions can affect quality of life. It is important to discuss quality of life with healthcare professionals because the professionals may provide helpful recommendations and support:

> 'First the GP and I talked about my situation. When we could not solve the problem, he referred me to a psychologist. I still talk to the psychologist. It was nice that he raised this topic because otherwise I might not have talked about it and I would not have a psychologist.' (female participant, 69 years in a focus group with a low level of education).

### Comprehensibility of TOPICS-MDS

Participants of Round 2 (n = 18) introduced different comments and improvements regarding the questions and response options of the TOPICS-MDS. As shown in Table 2, participants indicated that the questions were sometimes difficult to understand and therefore hard to answer. The most frequently mentioned reasons were the use of difficult words, complex questions, sensitive questions (e.g., about urinary incontinence), lengthy questions and a multitude

**Table 2. Comprehensibility of the questions (Qualitative data from the focus groups, Round 2).**

| Barriers to the understanding of questions | Quotes |
|---|---|
| **Language** | |
| A frequently mentioned barrier was the use of difficult words. In particular, participants with a culturally diverse background experienced difficulties understanding specific questions. | 'What is 'mood' (in Dutch 'stemming')? I do not know. Is 'stemming' if you hear voices in your head? (participant in a focus group; question 3). 'Yes, 'brain function' (in Dutch 'hersenfuncties'), that is a difficult word' (participant in a focus group; question 4). |
| Participants indicated that it would be helpful to provide examples when using a difficult word. | 'You may use the word 'incontinence products', but you can give examples like panty-liners, diapers or Tena Lady' (participant in a focus group; question 6). |
| Moreover, participants indicated that the use of abbreviations makes the question difficult to understand. | 'And all the abbreviations, you just have to know what it all means' (participant in a focus group; question 1). |
| **Formulation of the questions** | |
| Some questions can be interpreted in various ways, making them hard to answer. | **Question: Do you need help with taking your medicines?** 'There are actually two things. The first thing is which pills do I have to take in the morning and the second thing is how do I take the pills. Those are two different things. It is not clear what is meant by 'help'' (participant in a focus group; question 8). |
| **Nature of the questions** | |
| Participants also indicated that certain questions were too direct or too sensitive. They prefer not to answer those questions. | 'People are embarrassed. Yes, especially if a man comes into a room and the woman must ask the man about incontinence problems. He will be embarrassed' (participant in a focus group; question 6) |
| Participants have also mentioned that in several cultures the topic of 'incontinence' is taboo. Therefore, many persons do not talk about this topic. If they have to provide responses to this questionnaire, they would rather not answer this question. | 'Yes, (. . .) older people who get these questions need their children to help them to translate the questions. It may be that the children will skip these questions because of taboo or out of shame about asking their parents about this topic'(participant in a focus group; question 6). |
| **Length of the questions** | |
| Participants have indicated that some questions, especially the introduction of the question, were too extensive. Therefore, it was difficult to follow these questions. | 'You know that the last part of the sentence will be remembered. The information before the comma has already been forgotten. Therefore, it is important to make short sentences that are not divided into sections' (participant in a focus group; question 4). |
| **Multitude of topics** | |
| Participants of this study also noticed that different topics were asked within individual questions. | **Question: Please tick the box of the answer that best fits your situation. Brain functions such as memory, attention and thinking.** 'Ok, the first question. I have no problems with my memory, done. Do not ask the other things. That is too much in one question. You should not ask about all these topics at once' (participant in a focus group; question 4). |
| **High number of response options** | |
| Participants had difficulties answering specific questions due to a high number of response options. Participants proposed the use of a scale to visualize the response options. | 'Always, very often, quite often, sometimes, almost never, never. There are many different categories. I think that never and always are no good options, because everyone is nervous sometimes' (participant in a focus group; question 9). |

of TOPICS-MDS abbreviations within individual questions. In addition, participants mentioned that certain questions were difficult to answer due to the high number of response options (>5) or the lack of a specific answer (e.g., 'sometimes' instead of only 'yes / no'). Based on the comments in Round 2, the twelve questions were reformulated and evaluated by

participants in Round 3. Ten of the twelve restated questions were assessed as better than the original question. The restated questions were most positively rated when simple words were used. If difficult words were supplemented with examples for clarification, participants rated it also as 'clearer'. Furthermore, the use of a visual scale as a response style was evaluated as 'clearer'. Participants find a scale helpful since it clarifies the response options visually, which makes it easier to provide an answer.

## Discussion

The aim of this study was to explore whether the TOPICS-MDS could facilitate SDM conversations between healthcare professionals and older adults with MCC. The older adults who participated in this Delphi study evaluated the various health domains of the TOPICS-MDS in terms of importance to discuss the health domains when reflecting on one's preferred health outcomes. The older adults assessed the following original health domains of the TOPICS-MDS as being important to maintain: functional limitations, emotional wellbeing, social functioning and quality of life. The following domains were also added: coping with stress and dealing with health conditions and the effects on daily life. We found differences between older adults with a low level of education or a culturally diverse background and older adults with a high level of education. Although we have to be cautious in our interpretations due to the size of the panels, the older adults in our groups with a low level of education or a culturally diverse background more heavily stressed the importance of discussing the health domains with healthcare professionals than did the older adults with a high level of education.

The identified domains are consistent with the findings of earlier studies. For example, the importance of discussing functional limitations in healthcare communication has also been described in other studies on outcomes that are important to older adults [32–35]. For instance, Fried et al. (2002) emphasize the importance of functional outcomes for older adults when they consider a treatment [35].

In our study, older adults prefer to discuss emotional wellbeing with a healthcare professional. The older adults emphasized, however, the importance of a trustworthy relationship between the healthcare professional and the patient. These findings are in line with the results of the study by Ridd et al. (2009), who found that a long-lasting relationships and continuity of care are important when discussing sensitive topics [36]. Persons generally experience more superficial relationships with doctors and nurses in hospitals than with their general practitioner (GP), who they have known for a long time [36].

The results indicate that older adults find it important to discuss social functioning with their healthcare professionals. Participants indicated that they are likely to become lonely and discussing this topic with healthcare professionals can help them find solutions. These findings do not correspond with the findings of other studies; other studies have reported that lonely persons rarely discuss issues with their GP and that GPs typically ask persons about loneliness indirectly or not at all [37–39]. An explanation for this inconsistency with previous literature may be that participants in the present study described a hypothetical situation (i.e., if I get lonely, I think I would like to talk to my GP); however, most studies about loneliness report about experiences from people who are lonely. Kharichi et al. (2017) emphasized that a good relationship between healthcare professionals and patients is necessary when discussing sensitive topics such as loneliness [39].

An additional domain introduced by the participants themselves was to talk with healthcare professionals about how to cope with stress. Stress can cause physical complaints and mental problems that can impair persons in conducting everyday activities in life and the feeling of happiness [40]. However, we expect that similar to conversations on other sensitive topics, a

trusting relationship between healthcare professionals and patients is a prerequisite for conversations on stress.

The last domain introduced by participants was dealing with a chronic condition and the effects on daily life. Dealing with the chronic condition and the effects on life has many relations with self-management and adaptation to changes. Barlow et al. (2002) define self-management as 'a person's ability to manage the symptoms and the consequences of living with a chronic condition, including treatment, physical, social and lifestyle changes' [41]. A previous study about barriers to self-management identified poor communication with healthcare professionals as a barrier to active self-management [42]. Other studies have shown that healthcare professionals can support persons' self-management by educating persons and providing information about chronic conditions and how to manage them [43,44]. Although participants identified self-management as a separate domain from quality of life, we think there is a close relationship between these two domains [45]. Smith et al. (2008) introduced the Brief Resilience Scale to assess a person's resilience, defined as 'the ability to bounce back and recover from stress' [45]. It is worth investigating whether this scale could be added to the TOPICS-MDS to address these two domains, coping with stress and dealing with the chronic condition and the effects on daily life.

There may be some explanations for the differences found between older adults with a high level of education and older adults with a low level of education or a culturally diverse background. The findings of earlier studies support our idea that in the Netherlands, older adults with a low level of education or a culturally diverse background make more frequent use of healthcare resources, especially GP services, and generally have more chronic conditions than older adults with a high level of education [46,47]. This concept might imply that due to frequent contact with their GP, individuals feel confident discussing a variety of health domains. However, within different cultures, a variety of beliefs and values determine views on health and the need for healthcare services, ranging from the use of only family care to extensive use of professional care, indicating that there are many different circumstances that influence which topics older adults with a culturally diverse background wish to discuss with their doctor [48]. It would be interesting for future research to get a deeper insight in the differences between these specific groups of older adults.

With regard to the comprehensibility of the TOPICS-MDS in conversations, the most frequent comments regarding the questions were the use of difficult words, the formulation and length of questions, the sensitive nature of the questions and the multitude of topics within individual questions. These barriers were identified by both older adults with a high education level and older adults with a low education level or a culturally diverse background.

## Strengths and limitations

A strength of this study was the involvement of older adults themselves in assessing the comprehensibility of the TOPICS-MDS as a basis for conversations between older persons and healthcare professionals and the need to discuss topics with their healthcare professionals. Second, by the organization of both the focus group sessions and the online questionnaire, we ensured panel members with a low level of education and a culturally diverse background were included in groups who participated in online questionnaires infrequently [49,50]. However, the use of focus group sessions might have introduced slight bias since the opinions and ratings of others could have influenced the choices of participants, although the advantage of the sessions was that deep insights were gained. In addition, we did not involve participants with a culturally diverse background or a low level of education in Round 3 of the Delphi study, which influenced the credibility of the quantitative results. Another possible limitation

of this Delphi study was the loss of participants in the third round (second internet question-naire), which may have impacted the emerging consensus over time. Additionally, although panelists were committed to the study, a higher number of respondents and a better distribu-tion of participants among the groups may improve the robustness of the results. Further research is needed to adapt and validate the TOPICS-MDS according to the discussed recom-mendations and to enhance the cultural sensitivity of the TOPICS-MDS.

## Practice implications

The recommendations regarding the additional topics and the suggestions to improve the comprehensibility of the questions have been presented to the Dutch TOPICS Working Group. For the TOPICS-MDS to be used as input for individual healthcare conversations between older adults and healthcare professionals, the education of healthcare professionals is needed. Therefore, in the large scale SDM implementation program of the Dutch Care Insti-tute (Zorg Instituut Nederland), a specific project has been designed to improve SDM with the PROM TOPICS-SF (Short Form), supported by the Dutch Geriatric Society. As a part of this project health professionals at geriatric hospital wards are supported in the use of the results of this PROM in individual health care conversations. For example, through e-learning, they learn how to use the PROM in healthcare conversations. In addition, the Dutch seniors' orga-nisation KBO-PCOB and the Dutch migrant organization NOOM put a joint effort in the empowerment of older adults to prepare for a health care conversation with this PROM, with information sessions for older adults and educational material.

## Conclusion

This study yielded recommendations of older adults on how to use the TOPICS-MDS PROM in healthcare conversations. Older adults assessed the majority of the questions in the TOPICS-MDS as important for healthcare conversations and advised the inclusion of two more domains, namely, coping with stress and dealing with health conditions and the effects on life. Furthermore, the older adults identified barriers regarding the comprehensibility of the questions in the TOPICS-MDS. These recommendations benefit the discussion of pre-ferred health outcomes for older adults.

## Supporting information

**S1 File. Round 1: Internet questionnaire.**
(PDF)

**S2 File. Round 3: Internet questionnaire.**
(PDF)

## Author Contributions

**Conceptualization:** Ruth E. Pel-Littel, Cynthia S. Hofman, Mirella M. Minkman.

**Data curation:** Ruth E. Pel-Littel, Liesje Yu.

**Formal analysis:** Ruth E. Pel-Littel, Cynthia S. Hofman, Liesje Yu.

**Funding acquisition:** Cynthia S. Hofman, Mirella M. Minkman.

**Investigation:** Ruth E. Pel-Littel, Cynthia S. Hofman.

**Methodology:** Ruth E. Pel-Littel, Cynthia S. Hofman, Liesje Yu, Mirella M. Minkman.

**Project administration:** Ruth E. Pel-Littel.

**Supervision:** Cynthia S. Hofman.

**Validation:** Ruth E. Pel-Littel.

**Visualization:** Ruth E. Pel-Littel.

**Writing – original draft:** Ruth E. Pel-Littel.

**Writing – review & editing:** Cynthia S. Hofman, Silke F. Metzelthin, Franca H. Leeuwis, Jeanet W. Blom, B. M. Buurman, Mirella M. Minkman.

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
