## [Decision Letter · Decision Letter 0]

2 Jul 2019

PONE-D-19-16906

Recommendations of older adults on how to use the PROM ‘TOPICS-MDS’ in

healthcare conversations: a Delphi study

PLOS ONE

Dear Prof Peo-Littel,

Thank you for submitting your manuscript to PLOS ONE. After careful consideration, we feel that it has merit but does not fully meet PLOS ONE’s publication criteria as it currently stands. Therefore, we invite you to submit a revised version of the manuscript that addresses the points raised during the review process.

Please see comments below. 

We would appreciate receiving your revised manuscript by 01 September 2019. To enhance the reproducibility of your results, we recommend that if applicable you deposit your laboratory protocols in protocols.io, where a protocol can be assigned its own identifier (DOI) such that it can be cited independently in the future. For instructions see: http://journals.plos.org/plosone/s/submission-guidelines#loc-laboratory-protocols

We look forward to receiving your revised manuscript.

Kind regards,

Andrew Soundy

Academic Editor

PLOS ONE

Journal Requirements:

3. We note that you have stated that "The study protocol was approved by the institutional Review Board of the UMCU (19/355).". However, we understand from the documentation that you have provided that this study was not formally approved, but rather the requirement to obtain approval was waived by your ethics committee. We would be grateful if you could update the Ethics Statement and Methods section of your manuscript to indicate this.

Additional Editor Comments:

I have read the manuscript and I agree with the comments made by the reviewer. Please consider these comments very carefully and make it very clear why and if you dont make changes as a result.

Reviewers' comments:

Reviewer's Responses to Questions

**Comments to the Author**

1. Is the manuscript technically sound, and do the data support the conclusions?

Reviewer #1: Partly

2. Has the statistical analysis been performed appropriately and rigorously? 

Reviewer #1: Yes

3. Have the authors made all data underlying the findings in their manuscript fully available?

Reviewer #1: No

4. Is the manuscript presented in an intelligible fashion and written in standard English?

Reviewer #1: No

5. Review Comments to the Author

Reviewer #1: I commend the researchers / authors for their commitment to the health of older adults. However, there are a number of questions that need to be answered.

- the tool is titled for older adults and informal caregivers survey - these are two distinct population groups. Was there a difference in the findings?

- if this tool is useful then the need to educate health care professionals is important and needs to be addressed. How might this be conducted?- there is no discussion of the implications of this study for the education of health care professionals specific to the use of this tool.

- was this a Delphi study, or was it simply three focus groups with specific intentions? I would suggest the former as it does appear that a consensus was sought. It does not appear from the reported findings that a group consensus was achieved, as different objectives were identified (lines 126-131).

- focus groups are an effective strategy for some qualitative work but the nature of some of the group members may prevent the views of others from being heard. This is inconsistent with the Delphi process.

- how was the decision made regarding culturally diverse backgrounds and low education (lines 116-118; 134-135)? Define middle level participants please.

- the age of 50 appears quite a young old age and implies that all adults >50 have the same needs.

- line 235, what does "online panel" mean in table 1 and used in other lines eg. line 134, line 420?

- how is the inherent bias addressed if participants with culturally diverse backgrounds and low education were solely in focus group lines 134-135? This influences the credibility of the quantitative results.

- if the participants identified additional topics (lines 59-60) then is the topics tool actually useful? I would suggest perhaps not. How might the authors respond or acknowledge this point?

- how was the qualitative data actually analyzed? The authors identify that content analysis was conducted (line 56) but provide no description of the process.

- the term "patient" is often used e.g. lines 365, 370, however, the study was not done with patients. This is a limitation that is not explored by the authors.

6. PLOS authors have the option to publish the peer review history of their article (what does this mean?). If published, this will include your full peer review and any attached files.

Reviewer #1: No

---

## [Author Response · Author response to Decision Letter 0]

30 Aug 2019

I have incorporated all of your suggestions into my revision. They were very helpful, thank you. See the attachment 'response to reviewers' for full details about the reflections and revisions.

---

## [Decision Letter · Decision Letter 1]

12 Sep 2019

PONE-D-19-16906R1

Recommendations of older adults on how to use the PROM ‘TOPICS-MDS’ in healthcare conversations: a Delphi study

PLOS ONE

Dear  Prof Perl-Littel,

Thank you for submitting your manuscript to PLOS ONE. After careful consideration, we feel that it has merit but does not fully meet PLOS ONE’s publication criteria as it currently stands. Therefore, we invite you to submit a revised version of the manuscript that addresses the points raised during the review process.

I think it is getting there could you just answer my questions and update the manuscript. 

We would appreciate receiving your revised manuscript by 12 October 2019. To enhance the reproducibility of your results, we recommend that if applicable you deposit your laboratory protocols in protocols.io, where a protocol can be assigned its own identifier (DOI) such that it can be cited independently in the future. For instructions see: http://journals.plos.org/plosone/s/submission-guidelines#loc-laboratory-protocols

We look forward to receiving your revised manuscript.

Kind regards,

Andrew Soundy

Academic Editor

PLOS ONE

Additional Editor Comments (if provided):

Can I just check when I have done Delphi I have used quantative stats including Kendalls W and the McNemar test to verify items selected – I used a modified Delphi Okoli and Pawlowski, 2004 – but I wanted to check why you have not used any stats?

Also Qualitative section page 10 I would like some more details for the reader.

The reader will need to know your paradigmatic position if you are presenting qualitative data so please add this. E.g., for the purpose of the qualitative analysis we have positioned ourselves as subtle-realists. This is important because it tells the reader what you present the information as you do eg., for me as a subtle realist I may include numbers of people which identify with each statement or theme.

For the themes can you consider if there is any sub-themes

Line 292-332

Some readers will want to have a little more detail: e.g., when you mention limitations in daily life – what do you mean? I would expand the detail e.g.,

Some participants (x/x, xx%) wanted to discuss their limitations in daily life. Limitations included; (a) help understanding how to navigate the stairs (x/x, xx%), (b) assistance in preparing meals (x/x, xx%)

Think about a standard way to introduce the theme

identification of sub-themes focus on x most prevalent sub-themes

e.g.,

Functional limitations

This theme include X sub-themes: (a) the ability to walk to the shops – then include detail as above

-also quotes can more detail so in analysis say how you identify individuals e.g., individuals will be identified by gender, age and educational level (school S, university U, higher degree;HD) an example of this would be (PM49U; meaning male participant, 49 years, University level education)

This is just a suggestion but something like this given would just help the reader understand that the process was robust and considered. You could include something in a supplementary file instead just to reveal the process

Reviewers' comments:

Reviewer's Responses to Questions

**Comments to the Author**

1. If the authors have adequately addressed your comments raised in a previous round of review and you feel that this manuscript is now acceptable for publication, you may indicate that here to bypass the “Comments to the Author” section, enter your conflict of interest statement in the “Confidential to Editor” section, and submit your "Accept" recommendation.

Reviewer #1: All comments have been addressed

2. Is the manuscript technically sound, and do the data support the conclusions?

Reviewer #1: Yes

3. Has the statistical analysis been performed appropriately and rigorously? 

Reviewer #1: N/A

4. Have the authors made all data underlying the findings in their manuscript fully available?

Reviewer #1: Yes

5. Is the manuscript presented in an intelligible fashion and written in standard English?

Reviewer #1: Yes

6. Review Comments to the Author

Reviewer #1: While some English editing would be useful, the current version is acceptable. I would suggest that the authors be advised of the contribution that an English editor might make to the quality of a manuscript.

7. PLOS authors have the option to publish the peer review history of their article (what does this mean?). If published, this will include your full peer review and any attached files.

Reviewer #1: No

---

## [Author Response · Author response to Decision Letter 1]

29 Oct 2019

Thank you for reviewing our manuscript entitled 'Recommendations of older adults on how to use the PROM ‘TOPICS-MDS’ in healthcare conversations: a Delphi study' PONE-D-19-16906. We greatly appreciated the useful suggestions for improvement, and the opportunity to re-submit our manuscript. We have carefully studied all comments and suggestions, cumulating in a revised version of our manuscript. In the attachment 'repsonse to reviewers' you will find a point-by-point response.

---

## [Editor Report · Decision Letter 2]

4 Nov 2019

Recommendations of older adults on how to use the PROM ‘TOPICS-MDS’ in healthcare conversations: a Delphi study

PONE-D-19-16906R2

Dear Dr. Ruth Pel-Littel,

We are pleased to inform you that your manuscript has been judged scientifically suitable for publication and will be formally accepted for publication once it complies with all outstanding technical requirements.

With kind regards,

Andrew Soundy

Academic Editor

PLOS ONE

Additional Editor Comments (optional):

Thank you for making the updates to this manuscript.

Reviewers' comments:

The reviewer suggested acceptance of the previous version so the manuscript was not sent out to them.

---

## [Editor Report · Acceptance letter]

11 Nov 2019

PONE-D-19-16906R2 

Recommendations of older adults on how to use the PROM ‘TOPICS-MDS’ in healthcare conversations: a Delphi study 

Dear Dr. Pel-Littel:

I am pleased to inform you that your manuscript has been deemed suitable for publication in PLOS ONE. Congratulations! Your manuscript is now with our production department. 

With kind regards,

on behalf of

Dr. Andrew Soundy 

Academic Editor

PLOS ONE